# Image Feature Detectors in Agricultural Harvesting: An Evaluation

**DOI:** 10.3390/s23239497

**Published:** 2023-11-29

**Authors:** Zhihong Cui, Lizhang Xu, Yang Yu, Xiaoyu Chai, Qian Zhang, Peng Liu, Jinpeng Hu, Yang Li, Haiwen Chen

**Affiliations:** 1Agricultural Engineering School, Jiangsu University, Zhenjiang 212013, China; cuizhihong_cc@163.com (Z.C.); justxlz@ujs.edu.cn (L.X.); xfpaxy@ujs.edu.cn (X.C.); zhangq_jsu@ujs.edu.cn (Q.Z.); lp_ujs@126.com (P.L.); hujinpeng2018@gmail.com (J.H.); liyangjsu@163.com (Y.L.); 2Mechanical Engineering School, Jiangsu University, Zhenjiang 212013, China; haiwenchen1873@163.com

**Keywords:** feature detector, agricultural harvesting, evaluation

## Abstract

Image feature detection serves as the cornerstone for numerous vision applications, and it has found extensive use in agricultural harvesting. Nevertheless, determining the optimal feature extraction technique for a specific situation proves challenging, as the Ground Truth correlation between images is exceedingly elusive in harsh agricultural harvesting environments. In this study, we assemble and make publicly available the inaugural agricultural harvesting dataset, encompassing four crops: rice, corn and soybean, wheat, and rape. We develop an innovative Ground Truth-independent feature detector assessment approach that amalgamates efficiency, repeatability, and feature distribution. We examine eight distinct feature detectors and conduct a thorough evaluation using the amassed dataset. The empirical findings indicate that the FAST detector and ASLFeat yield the most exceptional performance in agricultural harvesting contexts. This evaluation establishes a trustworthy bedrock for the astute identification and application of feature extraction techniques in diverse crop reaping situations.

## 1. Introduction

Image features represent distinct regions within images, encapsulating specific information. They serve as the foundational step for an array of vision applications, including image matching, object recognition, SLAM (simultaneous localization and mapping), 3D reconstruction, and more. In recent years, image feature-based applications have gained significant traction in the agricultural sector [1,2,3,4,5,6,7,8]. In this context, the precision, effectiveness, and dependability of feature extraction are paramount. This not only augments the precision and quality of application performance and accelerates data processing, but also guarantees outstanding adaptability and stability within agricultural settings. However, due to the intricate and diverse nature of agricultural environments, selecting the most suitable feature extraction method for a specific scenario poses a highly challenging task.

Image feature detection methods have evolved over the past four decades. Initially, traditional hand-crafted detectors were employed, such as those based on first-order image information [9,10], second-order image differentials [11,12,13,14], or comparisons of pixel intensities between central points and their surroundings [15,16,17,18,19]. More recently, with advancements in deep learning techniques, numerous deep learning-based feature detectors have been proposed. Some are solely designed for feature detection [20,21,22,23,24], while others integrate feature detection into a matching pipeline [25,26,27,28,29]. In order to assess the efficacy of these feature detectors, extensive research has been conducted [30,31,32]. However, these evaluations rely on the Ground Truth relationship between images. In real-world agricultural harvesting scenarios, which often involve intense mechanical vibrations, copious amounts of dust, fluctuating illumination, and crop micro-movements due to wind, obtaining Ground Truth proves exceedingly difficult. Furthermore, there is a scarcity of publicly available datasets related to agricultural harvesting.

Addressing this shortfall, our study has compiled and made publicly available the first agricultural harvest dataset. This dataset encompasses four types of crops: rice, corn and soybean, wheat, and rape. Furthermore, we have devised a feature detector evaluation method that does not rely on Ground Truth. This approach comprehensively considers efficiency, reproducibility, and feature distribution as critical elements. Subsequently, using the collected dataset, we evaluated eight different feature detectors, namely FAST [17], AGAST [18], ORB [19], SIFT [11,12], SURF [13], Key.Net [24], SuperPoint [26], and ASLFeat [29], to explore the optimal feature extraction methods for different crops in agricultural harvesting scenarios. Empirical research findings indicate that the FAST detector [17] and ASLFeat [29] exhibit exceptional performance in the agricultural harvesting environments of these four crops. A schematic of our study’s methodology is depicted in Figure 1.

To the best of our knowledge, the realm of public datasets remains bereft of agricultural harvesting imagery collections, compounded by an absence of evaluating diverse feature detectors within this specific context. Consequently, this endeavor marks an inaugural exploration, evaluating the performance of an array of feature detectors under such uniquely challenging conditions. The principal contributions of this research can be encapsulated as follows:This study introduces the premier agricultural harvesting dataset, encompassing four distinct crops in mechanized harvesting settings.A thorough and Ground Truth-independent approach for appraising feature detector performance is devised, with eight diverse feature detectors assessed using the proposed agricultural harvesting dataset.The four-crop agricultural harvesting dataset (FCAHD) presented in this study is rendered publicly accessible at The four-crop agricultural harvesting dataset (kaggle.com)

The structure of the paper unfolds as follows: Section 2 offers a concise overview of the eight evaluated feature detection techniques. Section 3 delineates the proposed four-crop agricultural harvesting dataset, while Section 4 outlines the specific experimental implementation methods and presents the results. Finally, Section 5 concludes the paper.

## 2. Related Work

In this section, we will meticulously elaborate on the specific implementation of the eight feature extraction methods used in this academic work (FAST, AGAST, ORB, SIFT, SURF, Key.Net, SuperPoint, and ASLFeat), and summarize the advantages and drawbacks of each method based on empirical observations.

The FAST detector [14] constitutes a high-speed corner feature detector, targeting real-time frame-rate applications. To accomplish this, a machine learning algorithm—specifically, a decision tree [33]—is devised to classify all corners within training images, encapsulating the rules of the original FAST corner detector [34,35]. This decision tree is subsequently translated into C-code and employed as the corner detector. Additionally, non-maximal suppression (NMS) is implemented to address multiple adjacent features. This method boasts remarkable efficiency and repeatability, albeit at the expense of increased noise levels.

The AGAST detector [18] represents an enhanced iteration of the FAST detector [17]. In a bid to simultaneously bolster generality and performance, AGAST employs a binary decision tree in lieu of the ternary tree utilized by FAST. This binary decision tree is grounded in the principle of selecting a pixel to test and posing a question that can rapidly reduce entropy. Furthermore, AGAST alternates between multiple specialized trees, contingent upon alterations in pixel neighborhoods. Consequently, AGAST delivers high performance in arbitrary environments without necessitating additional training. However, its efficiency has diminished relative to that of FAST.

ORB [19] constitutes another notable enhancement of the FAST detector [17]. Within ORB, FAST corner features are identified in each layer of the scale pyramid constructed on images, and a Harris corner filter [10] is applied to arrange the features. Subsequently, the top N features are selected. Additionally, ORB computes the precise orientation of each chosen feature using intensity centroids [36]. As a result, ORB emerges as an efficient alternative to SIFT [11,12] or SURF [13].

SIFT [11,12] is among the most illustrious blob feature detectors. To identify distinct features, a Gaussian pyramid [37] is assembled on each image, and the Difference-of-Gaussians (DoG) operator [11] calculates local maxima (keypoints). Subsequently, a 3D quadratic function is fitted to the discrete keypoints, enabling more precise extreme point localization [38]. Additionally, a 2 × 2 Hessian matrix [39] eliminates unstable edge keypoints. Ultimately, a consistent orientation is assigned to each keypoint through a rotationally invariant measure [40]. The detected SIFT features demonstrate robust invariance to image scale, rotation, and limited illumination and affine variations.

SURF [13] functions as a speedier version of the SIFT detector [11,12]. To tackle the time-intensive issue posed by the DoG operator [11], SURF employs the determinant of the Hessian matrix [39] to compute local maxima, while a simpler box filter substitutes the Gaussian filter. Furthermore, image pyramids are constructed using box filters of varying sizes, rather than altering the dimensions of the original images. Consequently, SURF attains enhanced efficiency without compromising performance.

Key.Net [24] is a feature detector that amalgamates traditionally hand-crafted and learning-based CNN filters. The hand-crafted filters serve as anchor structures for the learning-based CNN filters, thereby reducing the overall number of parameters to be learned. The learning-based CNN filters are subsequently employed to locate, score, and rank the detected features. Additionally, a multi-scale pyramid is constructed to extract features at varying levels, and the loss function is designed to enhance the robustness and repeatability of the features. The results [24] indicate that Key.Net exhibits superior performance in feature detection.

SuperPoint [26] is a self-supervised framework capable of jointly extracting features and their corresponding descriptors. To facilitate self-supervised training, a base detector (MagicPoint) is pre-trained on a synthetic dataset (Synthetic Shapes) and subsequently utilized in conjunction with Homographic Adaptation to automatically label the target domain (Pseudo-Ground Truth). SuperPoint ingests the original full-sized image and employs a VGG-Style [41] encoder to reduce dimensionality before the feature decoder and descriptor decoder units. Consequently, SuperPoint demonstrates heightened efficiency and potential for practical applications.

ASLFeat [29] concurrently learns feature detectors and descriptors, striving to address two intrinsic limitations in this format, namely, the local shape (scale, orientation, etc.) and localization accuracy of the detected features. In ASLFeat, deformable convolutional networks (DCN) with geometric constraints are employed for enhanced dense estimation. Furthermore, more precise feature locations are acquired through an inherent feature hierarchy. Ultimately, more reliable detection scores are accessible via peakedness measurements. The experimental results [29] underscore the superiority and practicality of ASLFeat.

There is a wide variety of feature detectors, each with its own unique advantages and application scenarios, thereby rendering the precise selection of the most fitting feature detector for a specific operational milieu a formidable endeavor. This selection necessitates not only a comprehensive understanding of the technical details and performance attributes of various feature detectors but also an all-encompassing evaluation and comparison of their performance in different contexts.

Conventional methodologies for evaluating feature detectors predominantly stand on the pillar of direct comparisons with Ground Truth data, serving as the yardstick for the algorithms’ accuracy and robustness. These evaluations typically unfold within the controlled confines of laboratory settings, characterized by constant lighting conditions, rigid objects, and precisely controlled motion, often achieved by mechanisms like high-precision robotic arms. However, in certain environments, notably agricultural harvesting, the establishing of an accurate Ground Truth is extremely challenging. This is not only because it requires accurate annotation of a large number of images in a complex and challenging natural environment, but also because it is often difficult to achieve in practice.

Particularly within open farmland environments, where sunlight exposure changes all the time, and crops are easily displaced by wind or mechanical operations due to the flexibility of their structure. Furthermore, the intrinsic aspects of the farmland, be it standing crops, harvested remnants, or the soil itself, exhibit highly similar textural characteristics, which increases the difficulty of feature detection.

In light of these considerations, seeking a viable and effective feature detector evaluation strategy that does not depend on Ground Truth is of significant importance in exploring the optimal feature extraction methods within complex agricultural harvesting scenarios.

## 3. Materials and Methods

### 3.1. Data Collection

No public datasets exist pertaining to the agricultural harvesting operations of four major field crops—rice, corn and soybean, wheat, and rape—thus, this paper presents the Four-Crop Agricultural Harvesting Dataset (FCAHD).

To capture visual image data, a ZED stereo camera (Model: ZED 2i, Stereolabs Inc., San Francisco, CA, USA) is mounted on top of a combine harvester using an adjustable tilting device. The tilting angle, set at approximately 45 degrees, ensures the camera’s field of view remained focused on the crops, rather than on irrelevant backgrounds such as houses, trees, or the sky. The camera’s operation and the recording of all collected data are executed on an industrial personal computer (IPC), specifically the Model: MIC-7700, Asus, Taiwan, China, mounted at the base of the pilot cabin. Three different combine harvesters are employed—one each for rice, and corn and soybean, and one shared for wheat and rape. Figure 2 illustrates the camera and IPC deployment on the rice combine harvester, alongside examples of collected data.

Data collection for rice, wheat, and rape occurred in Zhenjiang, Jiangsu, China, while corn and soybean data collection took place in Yancheng, Jiangsu, China. Video data were collected via ZED SDK, amassing 57.4 GB of data over a 6-month period.

Rape: A distinct agricultural produce, rapeseed, known scientifically as Brassica napus, stands as a pivotal oilseed cultivation. The seeds, termed rapeseed, are primarily harnessed for the extraction of consumable oil and the formulation of livestock sustenance.

Corn and soybean: A corn-soybean intercropping model facilitates efficient sunlight utilization and yields an additional soybean season without reducing corn yields, widely promoted in China.

### 3.2. Data Pre-Processing

Data pre-processing involves the following steps:Step 1:Manually select suitable evaluation data for each crop, using a continuous linear harvesting benchmark that excludes discontinuous elements such as people, birds, or other harvesting machinery entering the frame.Step 2:Employ Stereolabs’ SVO Export to extract left and right views at 30-frame intervals.(https://github.com/stereolabs/zed-examples.git, accessed on 12 January 2023)Step 3:Utilize OpenCV to read all extracted images, resize them to a uniform 1280 × 720 resolution, rename them, and save them in folders organized by views and crops.


In general, FCAHD comprises a total of 860 images representing the four different crops in agricultural harvesting scenarios, including left and right views. The dataset includes 216 rice images, 270 corn and soybean images, 192 wheat images, and 182 rape images. The dataset is available online: the four-crop agricultural harvesting dataset (kaggle.com).

### 3.3. Ground Truth-Free Metrics

Traditional feature detector evaluation methods rely on direct comparison with known Ground Truth data to ascertain the algorithm’s accuracy and robustness. However, these data are extremely difficult to obtain in actual farmland environments. Therefore, instead of relying on direct comparisons with “real” data, we have opted to indirectly evaluate the performance of the feature detector by comprehensively analyzing its actual performance in this particular application scenario. The evaluative criteria we have instituted concentrate on the most critical elements of the application, such as the frequency of wrong matching and the feature stability, all quantifiable without tethering to a precise Ground Truth. Through this strategy, our approach provides a feasible and predictable route for feature detector evaluation in complex, real-world scenarios.

Following a comprehensive examination of prior correlated studies [30,31,32], we introduce a novel evaluation metric, noteworthy for its independence from Ground Truth data. For each cropped image in the dataset, feature points are first extracted using the method outlined in Section 2. The number of feature points and the associated computational time for each method are recorded. Thereafter, the correlations between identified image attributes are scrutinized in two dimensions: left-to-right and back-to-front viewpoints. In particular, left-to-right views refer to images from both left and right viewpoints captured by the stereo camera at the same moment. Back-to-front views are a series of images from the left viewpoint of the stereo camera captured continuously at a rate of 30 frames per second (30FPS). Feature descriptors and a matching technique serve as tools, and the respective repeatability rates are determined. Finally, the distribution of feature points and matching performance are appraised subjectively, and the final scores for each feature detection method for each crop are calculated.

To compare the performance of different feature detectors, three metrics are utilized: efficiency (EF), repeatability rate (RR), and final score (FS), as shown in Equations (1)–(3), respectively. Efficiency (EF) and repeatability rate (RR) are defined as:(1)$$EF={\left(\sqrt[{\lambda }]{\frac{1}{2n}\sum_{i=1}^{n}\frac{t_{L_{i}}}{N_{L_{i}}}+\frac{t_{R_{i}}}{N_{R_{i}}}}\right)}^{-1},$$
(2)$$RR=\frac{1}{n}\sum_{i=1}^{n}\frac{M_{{LR}_{i}}}{N_{L_{i+1}}+N_{R_{i}}}+\frac{1}{n-1}\sum_{i=1}^{n-1}\frac{M_{{BF}_{i}}}{N_{L_{i}}+N_{L_{i+1}}}\;,$$
where $N_{L_{i}}$ and $N_{R_{i}}$ represent the number of points extracted by each feature detector for each crop’s left and right views, respectively, while $t_{L_{i}}$ and $t_{R_{i}}$ denote the corresponding time consumption. λ signifies the number of root sings. $M_{{LR}_{i}}$ and $M_{{BF}_{i}}$ are the number of matches for the left-to-right and back-to-front views under each feature detector, respectively. n is the number of images for each crop in a single view. Conversely, the final score (FS) is given by:(3)$$FS=\omega_{1}\frac{EF-\mu_{EF}}{\sigma_{EF}}+\omega_{2}\frac{RR-\mu_{RR}}{\sigma_{RR}}+\omega_{3}\frac{AS-\mu_{AS}}{\sigma_{AS}}\;,$$
where AS is an artificial score based on the distribution of feature points and matching performance, with scoring benchmarks detailed in Section 3.4. $\mu_{EF}$, $\mu_{RR}$, and $\mu_{AS}$ are the mean values of efficiency, repeatability rate, and artificial score, respectively, while $\sigma_{EF}$, $\sigma_{RR}$, and $\sigma_{AS}$ represent the corresponding standard deviations. $\omega_{1}$, $\omega_{2}$, and $\omega_{3}$ denote the corresponding weights.

### 3.4. Artificial Score

To render the evaluation more comprehensive, we devised an artificial scoring standard that encompasses error matching and feature distribution. Figure 3 displays the metric. We establish 3 levels for error matching and feature distribution: Rank 1, Rank 2, and Rank 3, with corresponding scores of 15, 7.5, and 0, respectively. In terms of error matching, Rank 1 signifies excellent matching performance, with an incorrect matching rate below 7%. Rank 2 indicates an incorrect matching rate below 15%, while Rank 3 represents a mismatch rate exceeding 15%. Regarding feature distribution, Rank 1 implies that detected features are well-dispersed across the three primary semantic regions in agricultural harvest images (namely, crops, harvested areas, and reel). Rank 2 suggests a well-distributed arrangement over two semantic regions, while Rank 3 denotes that features are predominantly distributed in one semantic area or are unevenly distributed.

Based on the constructed criteria, we first manually distilled the matching error data for each feature detector, whence we established the error rank categories on the graph’s vertical axis. Thereafter, we executed pertinent selections on the horizontal axis, reflective of the disparities in the distribution performance of feature points. At the confluence of these axes, we specified the specific score of each method. This scoring method is unique in that it reveals the comparative strengths in practical applications even in the absence of Ground Truth, making it a flexible and practical evaluation strategy.

### 3.5. Implementation Details

The evaluation is conducted on a computer with an AMD Ryzen 9 3900X CPU, 16 GB of memory, and an NVIDIA GeForce RTX 2080Ti graphics card boasting 11 GB of video memory. The operating system is Ubuntu 18.04. For traditional hand-crafted methods [11,12,13,17,18,19,42], we utilize OpenCV 3 implementations. To maintain a reasonable number of feature points, we set the parameter threshold to 50 (default 10) for FAST and AGAST. In ORB, we set the parameter features to 4000 (default 500). For SIFT and SURF, we directly employ the original parameters in OpenCV 3. We implement deep learning-based feature detectors [24,26,29] using the code and pre-trained models provided by the authors. We set λ to 0.43 to minimize the efficiency gap. Integrating the unique characteristics of the dataset used in this study with the wisdom derived from prior research, we elected, following a comprehensive assessment of multifaceted considerations, to establish the weights as $\omega_{1}$ = 0.25, $\omega_{2}$ = 0.45, and $\omega_{3}$ = 0.3.

## 4. Results

### 4.1. Image Feature Detection and Efficiency

The extraction of image features and their corresponding efficiency are crucial metrics, particularly in agricultural applications with real-time requirements, such as image matching, crop recognition, and motion tracking.

Figure 4 illustrates examples of feature points extracted by various feature detectors in different crops. Figure 5 shows the continuous variation in the number of feature points extracted by each feature detector and their corresponding computation times under different crop conditions. Table 1 lists the average number of feature points in each view and their corresponding computational times. Table 2 displays the efficiency (EF) of each feature detector in each type of crop.

Upon meticulous analysis of the data presented in the above figures and tables, it becomes evident that diverse feature detectors display significant disparities in their efficacy. Concurrently, the effectiveness of a single feature detector fluctuates when applied to different crop types.

For instance, the FAST detector maintains a comparatively consistent time consumption throughout all crop varieties, demonstrating superior efficacy, especially within the image sequences of rice and rape. AGAST, albeit marginally more temporally demanding than FAST, outstrips in performance within the sequences for corn and soybean, and wheat. In contrast to FAST and AGAST, ORB incurs heightened time consumption, and its distribution is too concentrated to show a significant advantage. The SIFT detector, bearing the heaviest time consumption, emerges as incompatible with scenarios demanding expedited processing.

Regarding the triad of detectors, SUFR, Key.Net, and SuperPoint, their time consumption lies betwixt ORB and SIFT, striking a modicum of equilibrium. Specifically, SUFR exhibits commendable performance in sequences of rape image sequences. Conversely, ASLFeat, despite a time consumption similar to ORB, marginally underperforms while contending with sequences of wheat and rape.

In general, traditional hand-crafted methods surpass learning-based methods in terms of efficiency, with the FAST detector holding a distinct advantage and the AGAST detector coming in second. Among learning-based methods, SuperPoint and ASLFeat exhibit similar efficiency, both outperforming Key.Net. Furthermore, we discovered that the number of feature points extracted by the FAST and AGAST detectors varies considerably among different crops, particularly between corn, soybean, and rape, while other methods exhibit relatively minor differences.

Hence, it is crucial to choose the right feature detector when proceeding with a particular task. It is not only about processing speed, but also directly affects the success and efficiency of the task.

### 4.2. Image Matching and Repeatability Rate

Repeatability [30] is a fundamental attribute of feature detectors, reflecting the ability of a feature detector to recognize the same features in an image under varying conditions. However, most works [30,31,32] calculate repeatability based on the Ground Truth correspondence between images, which poses a significant challenge in real-world agricultural harvesting scenarios. In this evaluation, we approximate repeatability using feature description and matching methods, subsequently integrating it with a comprehensive manual assessment to score each feature detector. For traditional hand-crafted detectors, we employ the SIFT descriptor [11,12] due to its exceptional distinctiveness. Conversely, for learning-based feature detectors, given their tight integration of features and descriptors, we use their respective descriptors directly. FLANN [42] was uniformly employed as the matching method.

We perform image matching in two dimensions: Figure 6 displays left and right view matching, while Figure 7 illustrates back and front view matching (left viewpoint). Figure 8 presents the ongoing variation of repeatability rate (RR) for four distinct crops in left-to-right (LR) and back-to-front (BF) image matching. Table 3 lists the average repeatability rate for each feature detector matched per crop.

Generally, learning-based feature detectors outperform traditional manual detectors, although FAST and AGAST detectors still exhibit commendable performance. SuperPoint and ASLFeat yield the best results. Moreover, a considerable discrepancy may exist between the repeatability rate of left-to-right views and back-to-right views.

### 4.3. Artificial Score and Final Score

Table 4 presents the results of the manual scoring, highlighting the challenges faced in achieving well-distributed features and high-quality image matching in agricultural harvesting scenarios. Generally, ASLFeat performs the best, with SuperPoint, Key.Net, FAST, and AGAST also showing commendable performance. Finally, we calculate the Final Score (FS) and display it in Table 5 and Figure 9. Section 5 will discuss the comprehensive analysis of different feature detectors for each crop.

## 5. Discussion

Our research revealed that the complex structure of crops during the harvesting period causes different parts of the same plant (such as panicles, leaves, and stems) to display distinct imaging effects under uniform natural light, particularly when converting RGB images into grayscale. As a result, in traditional algorithms, methods that rely on pixel brightness comparisons (like FAST and AGAST) maintain an advantage in keypoint distribution and subsequent matching efficacy. Conversely, other methods, though capable of extracting deeper information, yielded less satisfactory results. Regarding learning-based techniques, performance variations primarily stemmed from differences in neural network architectures and the diversity of the overall training data. ASLFeat, which integrates attention mechanisms and self-supervised learning (ASL) alongside convolutional neural networks, attains exceptional generalization capacity, thereby achieving a distinct overall advantage.

After scrutinizing the summary tables derived from the preceding discussion, we discovered that learning-based feature detectors generally outperform traditional hand-crafted feature detectors, albeit at the cost of increased computational time. Among the traditional handcrafted feature detectors, the FAST detector exhibits the best performance due to its exceptional efficiency and considerable repeatability. While AGAST demonstrates nearly the same performance as FAST in terms of repeatability and feature distribution, its ability to adapt to diverse environments is not fully reflected in the harvesting scenarios of these four crops, and it also required more time. Additionally, ORB, SIFT, and SURF (original) are not recommended for use in agricultural harvesting scenarios.

Among the learning-based feature detectors, ASLFeat demonstrates the best overall performance. Key.Net and SuperPoint also perform well but are not entirely suitable for agricultural harvesting scenarios. In summary, for rice, we recommend using the FAST detector, SuperPoint, and ASLFeat. For corn and soybean, wheat, and rape, we suggest employing the FAST detector and ASLFeat.

It is noteworthy that due to the near impossibility of obtaining images with Ground Truth in the specific context of agricultural harvesting, we were unable to use images with true values for comparative analysis to demonstrate the efficacy of our designed evaluation method. However, precisely because of this, the metrics we proposed were meticulously crafted.

The first metric is the efficiency of feature extraction. In numerous applications, the computational efficiency of the entire system has been a focal point for developers, as the computational efficiency of feature extraction directly impacts the system’s response speed and practicality. Consequently, more efficient methods of feature extraction possess a significant advantage in subsequent applications.

Our second metric is repeatability. The location of important feature points often contains more valuable information, facilitating feature matching between images, with the results directly applicable to subsequent applications. Repeatability is a crucial measure of the ability to match feature points. If a feature extraction method typically extracts points of significant importance, rich in information, then the match relationships established by these points will also be richer, establishing a valid logical relation. Due to the lack of real information, we cannot directly judge the accuracy of the established matches, thus we introduced the rate of mismatches judged manually as our third metric. Therefore, methods that establish numerous matches with a lower rate of erroneous matches more accurately reflect the effectiveness of the feature extraction method in specific scenarios.

In our third metric, we also introduced an assessment of feature point distribution based on manual judgment. Feature points that are evenly distributed can cover the entire image, providing comprehensive information for scene understanding and global analysis. In tasks like SLAM (Simultaneous Localization and Mapping) or 3D reconstruction, an even distribution aids in more accurately estimating motion or reconstructing three-dimensional structures, reducing informational biases and concentrations. Additionally, uniform distribution also helps to minimize feature extraction biases under conditions of changing perspectives or uneven lighting. Therefore, the uniformity of feature point distribution will receive higher scores in the evaluation.

In summary, although we cannot directly validate the effectiveness of our metrics through agricultural harvest images with Ground Truth values, there is a strong positive correlation between the proposed evaluation metrics and the quality of the feature point extraction methods. The scores derived from our evaluation metrics are well-considered, reflecting their characteristics in agricultural harvesting scenarios.

## 6. Conclusions

Due to the scarcity of datasets and the difficulty in obtaining Ground Truth in adverse agricultural harvesting scenarios, selecting the optimal feature detector for specific scenes poses a significant challenge. In this study, we have collected and open-sourced the first agricultural harvesting dataset encompassing four crops: rice, corn and soybean, wheat, and rape. We have devised an innovative method for evaluating feature detectors without Ground Truth, thoroughly integrating efficiency, repeatability, and feature distributivity.

We evaluated eight distinct feature detectors on our collected Farming Crop Harvest Dataset (FCAHD). Experimental results reveal that the FAST detector excelled in efficiency, while ASLFeat demonstrates superior overall repeatability and distributivity. Moreover, we conducted a performance assessment of each algorithm’s efficacy across the four crop types, thus laying a solid foundation for future adoption of these algorithms. For rice, we recommend the FAST detector, SuperPoint, and ASLFeat. For corn and soybean, wheat, and rape, we suggest using the FAST detector and ASLFeat.

It is important to note that this paper focuses solely on the differences presented by various feature detectors and does not consider other potentially useful methods, such as descriptors and matching methods. Also, it is worth mentioning that the evaluation methodology proposed in this paper has a degree of subjectivity and does not fully replace conventional evaluative methods based on Ground Truth. It is more suitable for specific scenarios wherein Ground Truth proves elusive and traditional modes of assessment pose considerable challenges. In such circumstances, our approach can be considered a better alternative.

Looking ahead, our research trajectory is set to incorporate a more diverse array of feature detectors and endeavor to meld them with a wider range of descriptors and matching methods. Concurrently, we are also committed to further optimizing the proposed evaluation framework. Explicit avenues for enhancement might encompass the adoption of machine learning to reduce the subjective component ingrained in the evaluation process or delving into strategies for the synergistic integration of hypothetical data with some Ground Truth data, thereby improving the precision of the evaluation.

## Figures and Tables

**Figure 1 sensors-23-09497-f001:**
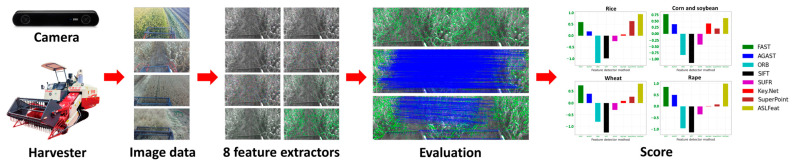
A visual portrayal of the research approach employed in this investigation. In the beginning, an image data procurement device is fastened to the mobile harvest platform, enabling the discrete gathering of harvest images for four individual crop species. Following this, a judiciously chosen assemblage of eight esteemed and symbolic feature extraction methods is examined by employing the evaluative framework devised in this treatise, obviating the necessity for Ground Truth. In the end, the performance of each algorithm relative to diverse crops is assessed and allocated a rating, establishing a reliable underpinning for the selection of algorithms in forthcoming implementations.

**Figure 2 sensors-23-09497-f002:**
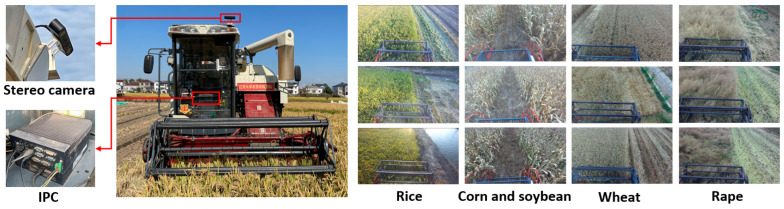
Camera and Industrial Personal Computer deployment, along with an example of the collected data.

**Figure 3 sensors-23-09497-f003:**
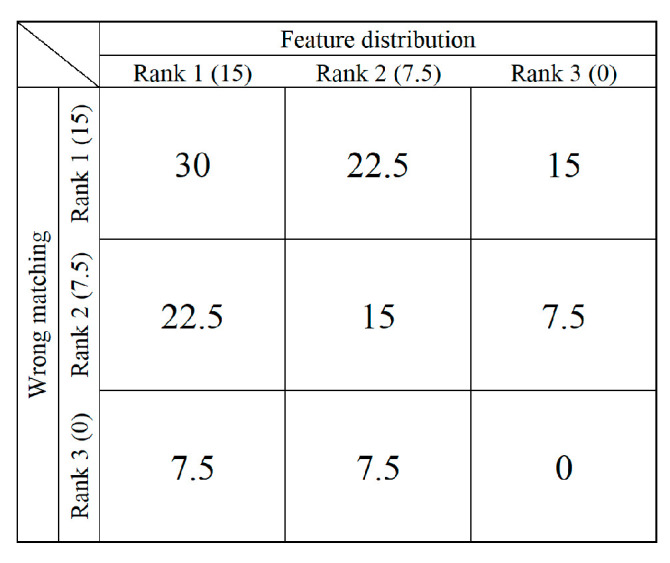
Artificial scoring criteria.

**Figure 4 sensors-23-09497-f004:**
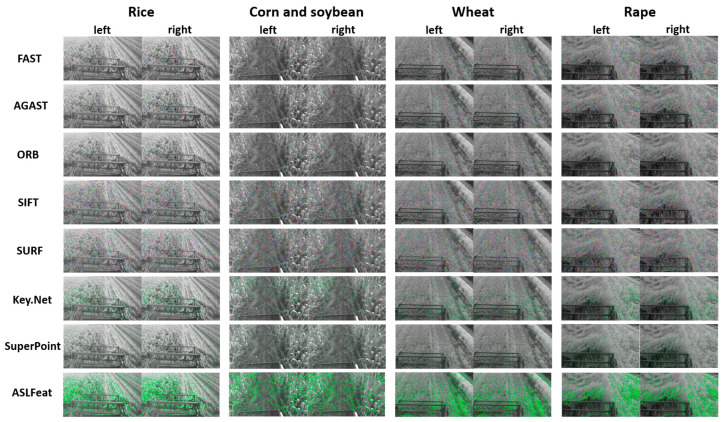
Feature point extraction by each detector across various crops. (In this manuscript, all images utilized for illustration and juxtaposition are accessible via the accompanying link. We cordially invite readers to download these images to obtain a more lucid and unobscured understanding of the original visuals).

**Figure 5 sensors-23-09497-f005:**
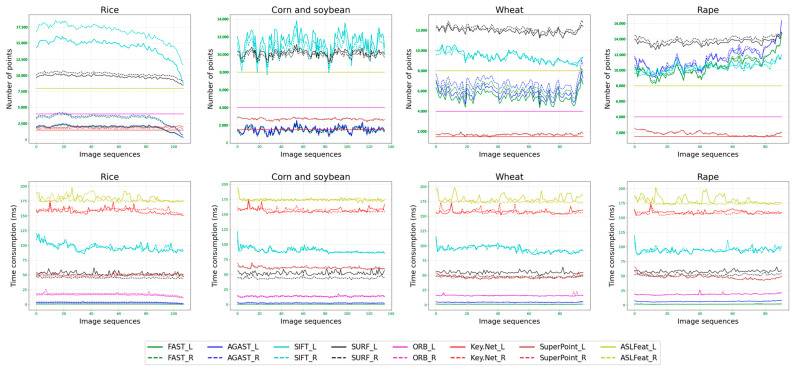
Ongoing fluctuation of feature points extracted by each detector in different crop scenarios and their corresponding time consumption. The solid line denotes the left view, while the dashed line represents the right view.

**Figure 6 sensors-23-09497-f006:**
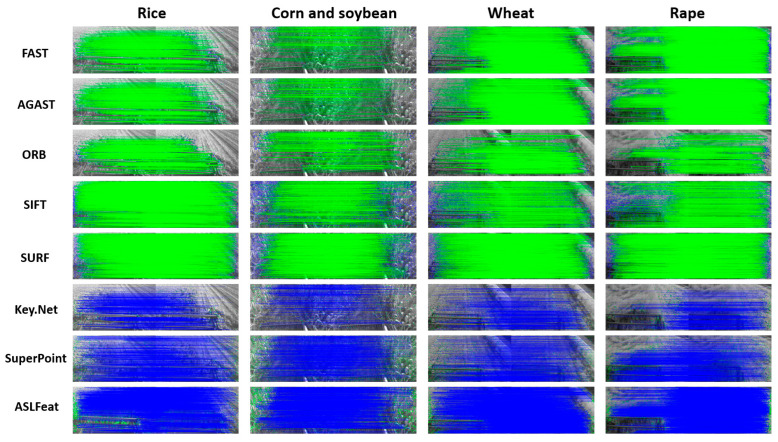
Matching between images from left and right viewpoints captured by the stereo camera at the same moment.

**Figure 7 sensors-23-09497-f007:**
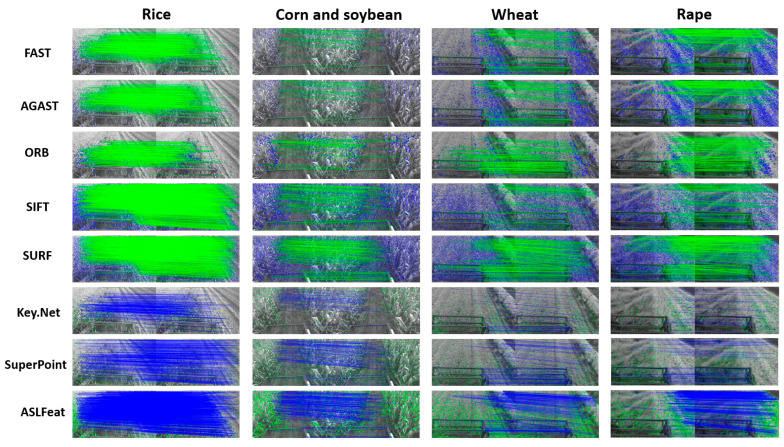
Matching between a series of images captured continuously at 30 frames per second (30FPS) from the left viewpoint.

**Figure 8 sensors-23-09497-f008:**
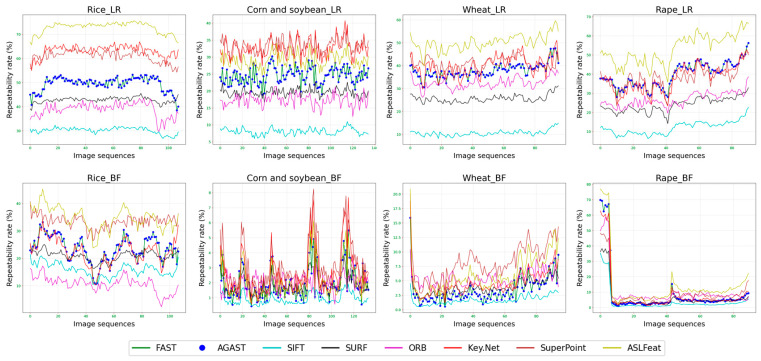
Sustained variation of repeatability rate (RR) across four distinct crops in left-to-right (LR) and back-to-front (BF) image matching.

**Figure 9 sensors-23-09497-f009:**
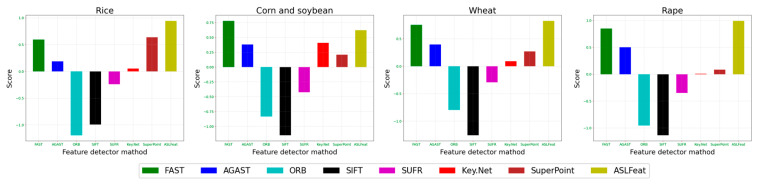
Final scores of each feature detector for individual crops (higher scores indicate superior performance).

**Table 1 sensors-23-09497-t001:** Average count of feature points for each detector matched per crop and associated computational time (time units are ms).

Detector	Crops							
	Rice		Corn and Soybean		Wheat		Rape	
	Left\Time	Right\Time	Left\Time	Right\Time	Left\Time	Right\Time	Left\Time	Right\Time
FAST	1857\0.66	3248\0.9	1488\0.69	1310\0.59	5380\1.15	6195\1.17	10,435\1.7	10,604\1.62
AGAST	1962\2.82	3428\3.85	1572\2.63	1387\2.35	5826\4.29	6714\4.56	11,442\6.38	11,624\6.28
ORB	4000\16.71	4000\18.0	4000\14.65	4000\12.99	4000\16.47	4000\15.96	4000\19.54	4000\18.48
SIFT	14,431\96.38	16,751\97.27	11,486\89.84	10,622\89.51	9390\94.21	9303\94.06	10,527\94.1	10,201\94.08
SUFR	9782\52.86	10,207\45.17	10,204\52.89	9904\44.11	11,917\55.65	12,164\47.85	13,568\58.65	13,978\51.7
Key.Net	1495\157.6	1459\160.2	1495\156.44	1459\156.72	1495\157.11	1459\157.27	1495\159.49	1459\157.72
SuperPoint	1964\50.47	1777\48.36	2683\61.35	2686\61.02	1707\47.11	1664\46.22	1832\48.41	1858\48.26
ASLFeat	8000\178.69	8000\177.86	8000\174.25	8000\174.43	8000\179.6	8000\174.38	8000\180.05	8000\174.79

**Table 2 sensors-23-09497-t002:** Efficiency (EF) of each feature detector for individual crops.

Detector	Efficiency			
	Rice	Corn and Soybean	Wheat	Rape
FAST	32.201	27.306	38.909	43.148
AGAST	17.648	15.585	22.639	25.238
ORB	10.380	11.459	10.682	9.979
SIFT	8.890	7.926	7.222	7.552
SUFR	9.875	9.941	10.445	10.758
Key.Net	2.608	2.625	2.620	2.610
SuperPoint	4.769	5.083	4.675	4.788
ASLFeat	5.133	5.182	5.149	5.144

**Table 3 sensors-23-09497-t003:** Average repeatability rate for each detector matched per crop. LR denotes left-to-right views, while BF signifies back-to-front views.

Detector	Descriptor	Matching	Repeatability Rate (%)							
			Rice		Corn and Soybean		Wheat		Rape	
			LR	BF	LR	BF	LR	BF	LR	BF
FAST	SIFT	FLANN	49.45	23.59	24.51	1.97	37.85	3.34	39.59	8.72
AGAST			49.45	23.59	24.51	1.97	37.85	3.34	39.59	8.72
ORB			38.93	10.66	17.27	2.00	32.40	5.37	26.73	10.51
SIFT			30.21	15.83	8.07	0.90	10.57	1.63	12.03	3.90
SUFR			42.79	20.97	19.57	1.82	25.61	3.86	24.04	6.27
Key.Net	Key.Net		63.12	22.48	32.63	2.69	40.20	4.61	37.81	8.75
SuperPoint	SuperPoint		60.96	33.48	33.34	3.16	40.34	8.15	38.06	9.56
ASLFeat	ASLFeat		72.81	34.65	29.10	2.06	50.11	6.01	53.14	14.39

**Table 4 sensors-23-09497-t004:** Artificial scores of each feature detector matched per crop.

Detector	Artificial Score			
	Rice	Corn and Soybean	Wheat	Rape
FAST	22.5	22.5	22.5	22.5
AGAST	22.5	22.5	22.5	22.5
ORB	7.5	7.5	0	0
SIFT	15	15	15	15
SUFR	22.5	15	22.5	22.5
Key.Net	22.5	22.5	22.5	22.5
SuperPoint	30	15	22.5	22.5
ASLFeat	30	30	30	30

**Table 5 sensors-23-09497-t005:** Final scores of each feature detector matched for individual crops. A more visually intuitive presentation of the results can be found in Figure 9.

Detector	Final Score			
	Rice	Corn and Soybean	Wheat	Rape
FAST	0.595	0.779	0.754	0.848
AGAST	0.189	0.382	0.399	0.502
ORB	−1.192	−0.831	−0.796	−0.953
SIFT	−0.99	−1.149	−1.257	−1.135
SUFR	−0.238	−0.424	−0.291	−0.348
Key.Net	0.054	0.41	0.093	0.009
SuperPoint	0.638	0.21	0.271	0.085
ASLFeat	0.943	0.624	0.827	0.991

## Data Availability

The dataset can be accessed at: https://pan.baidu.com/s/1mt38fzQ0sQiT9-RGL5PByA?pwd=mtzb and the four-crop agricultural harvesting dataset (kaggle.com), accessed on 1 November 2023).

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
