# Peer review of "Image Feature Detectors in Agricultural Harvesting: An Evaluation"

_sensors, 2023, doi:10.3390/s23239497_

Round 1
Reviewer 1 Report
Comments and Suggestions for Authors
The main proposed approach has novelty in contribution and methodology. Revision in terms of technical details is needed before publication. Also, paper organization can be improved. In this respect, some comments are suggested to describe technical details.
1. Discuss about the reported results in the Figure 5 with more details. Discuss about the obtained goals using this figure.
2. Why do you use 30 frames to train the model? Is the performance of the proposed approach sensitive to the number of frames?
3. How do you select the scores in the Figure 3?
4. Related works is a main section in scientific papers. So, it is better to review more related papers about feature extraction in agricultural images. For example, I find a paper titled “Bark texture classification using improved local ternary patterns and multilayer neural network”, which has enough relation. Cite this paper and some other related.
5. Did you implement all of the compared methods in the Table 5? If no, add related references.
Reviewer 2 Report
Comments and Suggestions for Authors
This work presents an novel approach of feature detector assessment and examine evarious feature detectors in this metric. It is encouraging for to make publicly available the inaugural agricultural harvesting dataset for related researchers.
1. The vertical axis in the last part of Figure 1 regarding Score is ambiguous. Please indicate it in Figure 1 or explain it in text.
2. In the introduction section of the article, the author only introduced the methods of image feature detection methods, and it is necessary to introduce some of the latest domestic and international achievements in image feature detection methods for agricultural harvesting.
3. In Chapter 2, it is recommended to summarize and explain the common problems of various feature extraction methods and the related reasons for proposing the novel assessment metric.
4. Please clarify the theoretical mechanism and purpose of this ground truth-free metric in section 3.3 instead of simply listing the implementation process.
5. Please unify the writing of variables in the article, and use the same mathematical symbols to represent the same variable clearly. For example, the writing format of formulas (1) and (2) is different from the text in the next paragraph, and the variable \lambda is expressed undefinedly in Section 3.5.
6. The expression "left to right and back to front viewpoints" mentioned in Section 3.3 are not clearly represented in Figure 4 , 6 and 7. What are the blue and green contents in Figure 6 and 7?
7. Other image precossing methods in feature detection should be addressed in the literature review part, e.g., active contour model, Active contour model based on local Kulback-Leibler divergence for fast image segmentation, Engineering Applications of Artificial Intelligence 123, 106472.
Comments on the Quality of English LanguageMinor editing of English language is required.
Reviewer 3 Report
Comments and Suggestions for Authors
This paper investigated the performance of eight common image feature detectors/descriptors in agricultural harvesting scenarios. Since it is difficult to annotate ground truth in the harvesting scene images, this paper defined three evaluation indices that did not rely on the ground truth. From the comparison, this paper concluded that FAST and ASLFeat detectors outperformed the others in agricultural harvesting scenarios.
However, since the correspondence determined by the distance of descriptors is not a “true” correspondence, the evaluation that this paper performed cannot be said to be a correct evaluation unless the ground truth is created and compared in some way, as in previous studies. The previous studies have also worked hard to create ground truth because they know this fact.
Though the priority is to correct the evaluation methods, I will describe other points for improving the paper’s quality. First, it is not good that the evaluation of "artificial scoring criteria" is subjective rather than algorithmic. This is not a fair comparison in this case. Second, if the wrong matching rate could be calculated, why would not you use it directly, rather than the ranks obtained by thresholding? If this is better in the evaluation, you should describe why. Third, you should describe the manner of determining the hyperparameters in Section 3.5. The evaluation should be changed by changing these parameters.
I will describe a few more points that caught my attention. In Equation (1), why did you apply the additional nonlinear function (e.g. root and inverse)? In Equation (2), why did you not add the repeatability rate of the right images’ back-to-front values? Is N_{L_{i+1}} in the first term a typo of N_{L_{i}}? Please correct \lambda in Line 232.
Comments on the Quality of English LanguageSentence phrasing and words are unique and difficult to read. Is "rape" a typo of "grape"?
Round 2
Reviewer 1 Report
Comments and Suggestions for Authors
Most of comments are considered by authors in the revised version. Added descriptions to the text, describe the existing methods in a more clear way. No new comment is suggested.
Reviewer 2 Report
Comments and Suggestions for Authors
The authors have addressed all my comments.
Comments on the Quality of English LanguageMinor editing of English language is required.
